# The Multiomics Response of *Bacillus subtilis* to Simultaneous Genetic and Environmental Perturbations

**DOI:** 10.3390/microorganisms11081949

**Published:** 2023-07-30

**Authors:** Li Liu, Gaoyang Li, Huansheng Cao

**Affiliations:** Division of Natural and Applied Sciences, Duke Kunshan University, Suzhou 215316, China; li.liu@dukekunshan.edu.cn (L.L.); lgyzngc@gmail.com (G.L.)

**Keywords:** *Bacillus subtilis*, bioengineering, *N*-acetylglucosamine, metabolome, transcriptome, systems biology

## Abstract

How bacteria respond at the systems level to both genetic and environmental perturbations imposed at the same time is one fundamental yet open question in biology. Bioengineering or synthetic biology provides an ideal system for studying such responses, as engineered strains always have genetic changes as opposed to wildtypes and are grown in conditions which often change during growth for maximal yield of desired products. So, engineered strains were used to address the outstanding question. Two *Bacillus subtilis* strains (MT1 and MT2) were created previously for the overproduction of *N*-acetylglucosamine (GlcNAc), which were grown in an environment with a carbon shift from glucose to glucose and xylose in the same culture system. We had four groups: (1) a wildtype (WT) grown with glucose at t1; (2) a WT with glucose and xylose at t2; (3) a mutant (MT1) grown with glucose at t1; and (4) MT1 with glucose and xylose at t2. By measuring transcriptomes and metabolomes, we found that GlcNAc-producing mutants, particularly MT2, had a higher yield of *N*-acetylglucosamine than WT but displayed a smaller maximum growth rate than the wildtype, despite MT1 reaching higher carrying capacity. Underlying the observed growth, the engineered pathways leading to *N*-acetylglucosamine had both higher gene expression and associated metabolite concentrations in MT1 than WT at both t1 and t2; in bioenergetics, there was higher energy supply in terms of ATP and GTP, with the energy state metric higher in MT1 than WT at both timepoints. Additionally, most top key precursor metabolites were equally abundant in MT1 and WT at either timepoints. Besides that, one prominent feature was the high consistency between transcriptomics and metabolomics in revealing the response. First, both metabolomes and transcriptomes revealed the same PCA clusters of the four groups. Second, we found that the important functions enriched both by metabolomes and transcriptomes overlapped, such as amino acid metabolism and ABC transport. Strikingly, these functions overlapped those enriched by the genes showing a high (positive or negative) correlation with metabolites. Furthermore, these functions also overlapped the enriched KEGG pathways identified using weighted gene coexpression network analysis. All these findings suggest that the responses to simultaneous genetic and environmental perturbations are well coordinated at the metabolic and transcriptional levels: they rely heavily on bioenergetics, but core metabolism does not differ much, while amino acid metabolism and ABC transport are important. This serves as a design guide for bioengineering, synthetic biology, and systems biology.

## 1. Introduction

Bacteria face multiple stresses at the same time. But how they respond at a systems level to simultaneous genetic and environmental perturbations is one fundamentally important yet open question in biology. Multiomics approaches have shown tremendous power in revealing the coordinated systems biology responses to perturbations [1]. This integrated approach can not only reveal the cellular state of bioenergetics, redox balance, and metabolism, but also demonstrate the coordinated operations of entire cellular system in response to perturbations [2,3].

However, most studies so far have relied on multiomics applied to microorganisms growing in only one of the two factors—internal (genetic) and external (environmental) factors. For example, *Bacillus subtilis* was grown at 37 °C, 45 °C, and 53 °C, addressing the response to the factor of the external environment but not genetic factors [4], and multiomics analyses of the mechanism for the formation of soy-sauce-like and soybean flavor in *B. subtilis* were carried out [3,4]. On the other hand, other studies of multiomics have usually studied the microorganisms of many genetic backgrounds but in the same environment. For example, the evolutionary trajectories of *Escherichia coli* in utilizing a toxic compound, fluorinated indoles, have been studied using multiomics [5]. Despite missing one or another factor, these integrated multiomics approaches have shown great power in probing biology and new knowledge has been gained. For example, in the experimental evolution of *E. coli* toward using fluorinated indoles (4- or 5-fluoroindole), few mutations arise to provide adaptation to both fluorinated Trp analogues, which are accompanied by a global adjustment of transcriptional regulation, membrane integrity, and quality control in protein folding. These findings highlight that only a few mutations are needed for bacteria to utilize unnatural amino acids, providing a systems biology foundation for bioengineering novel microbial strains for synthetic biology.

Bioengineering or synthetic biology provides an ideal system for studying the systems biology responses to simultaneous genetic and environmental perturbations [6]. For example, a *B. subtilis* strain was engineered for the mass production of *N*-acetylglucosamine, which has genetic changes (the genes *GNA1* and *glmS* are introduced and overexpressed) and environmental changes (i.e., grown first with glucose, then with both glucose and xylose (also an inducer of gene overexpression)) [7]. Importantly, a shift in the carbon nutrients occurs in the same flasks at steady state, i.e., the middle exponential phase.

In this study, we investigated the systems biology response of *B. subtilis* to simultaneous genetic and environmental changes. We recreated the *N*-acetylglucosamine-producing *B. subtilis* strain used previously and grew it and the wildtype, each in minimal medium first with glucose, then with glucose and xylose. Both transcriptomic and metabolomic changes were measured to achieve systems-level responses to simultaneous genetic and environmental perturbations.

## 2. Methods

### 2.1. Strains and Growth Conditions

*Bacillus subtilis* subsp. *Subtilis* str. 168 was the wildtype strain. All *B. subtilis* strains (Table 1) used and created in this study were grown at 37 °C in Luria–Bertani broth (LB, 10 g/L tryptone, 10 g/L NaCl, and 5 g/L yeast extract). The minimal medium with glucose as the sole carbon source contained (g/L) the following: Na_2_HPO_4_ 7.1, KH_2_PO_4_ 1.35, (NH_4_)_2_SO_4_ 2, MgSO_4_·7H_2_O 0.514, FeSO_4_·7H_2_O 0.01, MnSO_4_·H_2_O 0.076, thymine 0.01, tryptophan 0.01, and glucose 2.0. Xylose was added up to a final concentration of 5 g/L when the OD_600_ reached 0.4.

### 2.2. Construction of Plasmids and Transformation

Plasmids p7Z6, pSTC, and pP43NMK were provided by Dr. Long Liu at the Key Laboratory of Carbohydrate Chemistry and Biotechnology, Jiangnan University. Following the methods by Liu Lab [7], the two strains which can produce GlcNAc were recreated. Specifically, the plasmids pP43-GNA1 and pSTOP-GNA1-glmS were first recreated following their methods using the three plasmids. Next, the wildtype *B. subtilis* 168 was transformed with each of the plasmids pP43-GNA1 and pSTOP-GNA1-glmS with a Bio-Rad MicroPulser Electroporator (Hercules, CA, USA) to obtain two engineered strains, respectively (Table 1).

### 2.3. Measurement of Growth and Sampling

The strains were revived from glycerol stock in 10 ml of LB medium and grown overnight at 37 °C and 120 rpm. Cells were then accustomed to the minimal medium by allowing them to complete two growth cycles in it. For the tests, four groups were set up: (1) WT grown with glucose at t1 (timepoint 1); (2) WT with glucose and xylose at t2 (timepoint 2); (3) MT1 grown with glucose at t1; and (4) MT1 with glucose and xylose at t2. Specifically, in the first two acclimation cycles, 100 µL of overnight culture was added to 10 mL of minimal medium to grow at 37 °C and 120 rpm. When OD_600_ reached 0.4, xylose was added to the medium to a final concentration of 5 g/L. Cells at the stationary phase were taken to minimal medium to start a second cycle. The samples were collected in the third growth cycle; 10 mL of cells was harvested at OD_600_ 0.4 at t1 from a 200 mL culture in a 500 mL Erlenmeyer flask. The cells were centrifuged at 6000 rpm for 5 min and the pellets were frozen in liquid nitrogen and stored at −80 °C. After xylose addition to a final concentration of 5 g/L, the cells were allowed to grow to OD_600_ 0.6 (t2) and 10 mL of culture was collected and processed as in t1.

### 2.4. Metabolomics Analysis

*Metabolite extraction.* An aliquot (100 mg) of samples was extracted on ice with 800 μL of cold methanol/acetonitrile/water (2:2:1, *v*/*v*) solvent with mixing. To perform the absolute quantification of metabolites, stock solutions of stable-isotope internal standards were added to the extraction solvent at the same time. The samples were then shaken vigorously for 2 min, incubated on ice for 20 min, and centrifuged at 14,000× *g* for 20 min. The supernatant was filtered through a 96-well protein precipitation plate, and then, the elution was collected and dried in a vacuum centrifuge at 4 °C.

*LC-MS/MS detection.* The samples prepared above were redissolved in 100 μL of acetonitrile/water (1:1, *v*/*v*) solvent. Detection was performed on an ultraperformance liquid chromatograph (1290 Infinity LC, Agilent Technologies, Santa Clara, CA, USA) coupled with QTRAP MS (AB 6500+, AB Sciex). The analytes were separated on HILIC (Waters UPLC BEH Amide column, 2.1 mm × 100 mm, 1.7 µm) and C18 columns (Waters UPLC BEH C18-2.1 × 100 mm, 1.7 μm). For HILIC separation, the column temperature was set to 35 ℃ and the injection volume was 2 μL. Mobile phase A was: 90% H_2_O + 2 mM ammonium formate + 10% acetonitrile; mobile phase B was: 0.4% formic acid in acetonitrile. The gradient was 85% B at 0–1 min, 80% B at 3–4 min, 70% B at 6 min, 50% B at 10–15.5 min, and 85% B at 15.6–23 min. It was then initiated at a flow rate of 300 μL/min. For RPLC separation, the column temperature was 40 °C and the injection volume was 2 μL. Mobile phase A was: 5 mM ammonium acetate in water; mobile phase B was: 99.5% acetonitrile. The gradient was 5% B at 0 min, 60% B at 5 min, 100% B at 11–13 min, and 5% B at 13.1–16 min. It was then initiated at a flow rate of 400 μL/min.

Positive and negative polarity switch modes were employed for the 6500+ QTRAP (AB SCIEX) instrument. The source conditions for ESI positive mode were as follows: a source temperature of 580℃, Ion Source Gas1 (GS1) at 45, Ion Source Gas2 (GS2) at 60, Curtain Gas (CUR) at 35, and IonSpray Voltage (IS) at +4500 V. The source conditions for ESI negative mode were as follows: a source temperature of 580 °C, Ion Source Gas1 (GS1) at 45, Ion Source Gas2 (GS2) at 60, curtain gas (CUR) at 35, and IonSpray Voltage (IS) at −4500 V. The acquisition of mass spectrometry quantitative data was carried out using the multiple reaction monitoring (MRM) method. Quality control (QC) samples were interspersed within the sample queue to assess the stability and reproducibility of the system.

*Data processing.* Quantitative data processing was performed using either MultiQuant or Analyst software. Quality control (QC) samples were processed alongside the biological samples. Metabolites in the QC samples exhibiting a coefficient of variation (CV) of less than 30% were considered to be reproducible measurements.

*Statistical analysis.* The processed data underwent sum normalization and were subsequently imported into SIMCA-P (version 14.1, Umetrics, Umea, Sweden) for multivariate data analysis, which included Pareto-scaled principal component analysis (PCA) and orthogonal partial least-squares discriminant analysis (OPLS-DA). To assess the robustness of the model, 7-fold cross-validation and response permutation testing were employed. The variable importance in the projection (VIP) value of each variable in the OPLS-DA model was calculated to determine its contribution to the classification. Statistical significance was determined using an unpaired Student’s t-test, with a *p*-value of less than 0.05 considered to be significant.

### 2.5. Transcriptional Analysis

Total RNA was extracted from the samples using a Qiagen RNeasy Micro Kit (Germantown, MD, USA) following the manufacturer’s instructions. The concentration and purity of the extracted RNA were examined on a Nanodrop 2000 (Waltham, MA, USA), and the RNA integrity was examined using agarose gel electrophoresis. RIN values were determined by Agilent 2100 (Agilent Technologies, Santa Clara, CA, USA). After the rRNA was removed and mRNA was fragmented, six-base random primers (random hexamers) were added to synthesize one-stranded cDNA with reverse transcriptase, with mRNA as the template, followed by two-stranded synthesis to form a stable double-stranded structure. After library preparation, sequencing was performed on an Illumina Hiseq 2500 platform. These raw data were deposited onto the NCBI Short Read Archive (SRA) database with a project accession PRJNA954209. Raw reads were quality-controlled using FASTQC and trimmed using Trimmomatic 0.39 [8] with a quality score of 26. The read counts for each gene were analyzed using RSEM [9]. Differentially expressed genes were identified using DESeq2 [10] and enriched GO (gene ontology) and KEGG terms were obtained through the webserver DAVID [11].

### 2.6. Correlation between Transcriptome and Metabolome

The correlation between transcriptome and metabolome was analyzed using two approaches. First, direct correlations between them were calculated using the R [12] base function *cor*. The genes showing high correlations with metabolomes were extracted to enrich GO and KEGG terms in the webserver DAVID [11]. Co-occurrence network analysis was performed using R package WGCNA [13] to identify the groups of the genes associated with metabolites. The modules (coexpressed subnetworks) were also used to enrich GO and KEGG biological functions using DAVIDe.

## 3. Results

The systems biology responses of GlcNAc-producing *B. subtilis* to genetic and environmental perturbation were addressed through five types of analyses below: gene expression and metabolite abundance in the engineered pathways, global metabolomic change, global transcriptional change, PCA clustering by metabolomics and transcriptomics, correlation between genes with metabolites, and the correlation of the modules of genes (from coexpression network analysis) with metabolites.

### 3.1. Systems Biology Responses in Central Metabolism with GlcNAc Production

The growth of the WT and the engineered MT1 (with only GNA1 introduced) (Figure 1A) and MT2 (with both *glmS* and GNA1 introduced) was first measured in minimal medium. Although smaller growth rates were observed for MT1, it reached a higher carrying capacity than WT (Figure 1B). In terms of yield at the stationary phase, MT2 showed the highest yield released into medium under aerobic conditions among all the strains at both timepoints (Figure 1C). Under anaerobic conditions, the yields were about 14.17-fold and 40.65-fold higher for both MT1 and MT2 under aerobic conditions, respectively (Appendix A). Next, the systems biology responses in each group in central metabolism were examined. Strikingly, the gene expression and metabolites displayed similar patterns. First, we observed a high expression of GNA1 by 46% in MT1 and an increase in intracellular GlcNAc production of more than 128%.

Most genes in central metabolism were more highly expressed at time 2 than time 1 in either MT or WT strain; the metabolites associated with these genes mostly also displayed a similar pattern: higher at time 2 than 1. Meanwhile, some metabolites were higher in MT2 than any other group, e.g., glucose 6-P, glucosamine 6-phosphate, and N-acetylglucosamine 6-phosphate.

### 3.2. Global Metabolomic Changes between Four Growth Conditions

A total of 278 metabolites of 37 categories were detected using targeted profiling (Figure 2), with carboxylic acids and derivatives having the most (66; 34%) metabolites, followed by organooxygen compounds (39; 14%) and fatty acyls (36; 13%). Two types of metabolomes were compared: (1) some key metabolites in energetics, redox, and essential precursors, and (2) metabolites being differently abundant between the treatments.

For the first type of comparison, we examined the cellular state with three types of indicator metabolites: energetics, redox, and essential precursors between the four groups. First, most interestingly, the energy state metric ranged from 0.443 to 0.496 and was not different between the two strains at either timepoint or between both timepoints in either strain (Figure 2A), while all the components of the metric (i.e., ATP, ADP, and AMP) were higher at time 2 than 1 in either WT or MT1, with ADP being the most abundant (Figure 2B–D). A second energy compound, GTP, showed similar patterns (Figure 2E–G). The contents of these six metabolites were all higher in MT1 than WT at both timepoints (*t* test, *p* < 0.05). Second, we also assessed the redox state of the cells in terms of the NAD^+^/NADH ratio. The ratio was significantly lower in WT (0.218) than MT1 (0.422), while the other three groups were not significantly different from each other, with WT being 0.292 and MT1 0.263 at time 2. Moreover, two other redox compounds, NADP and FAD, also showed similar patterns to those of NAD and NADH (Figure 2H–K). Third, we examined the concentrations of 13 of the top essential precursors (Figure 2L–X). Most of them showed two patterns. In WT, most of them were higher at time 2 than time 1; in MT1, most of them were also higher at time 2 but some were, at time 2, equal to or lower than time 1, such as acetyl-CoA (Figure 2Q) and glutamine (Figure 2U).

Among the 278 metabolites in the targeted profiling, there were many more (not fewer) metabolites which were more abundant in WT at t2 (72 metabolites) than (6), WT (54) than MT1 (17) at t1, MT1 at t2 (123) than t1 (7), and MT1 at t1 (40) than WT at t2 (34) (Figure 3). Among them, there were 13 metabolites which were always differently abundant between any of the four groups: 5’-Deoxyadenosine, Adenosine 3’,5’-diphosphate, Guanosine monophosphate (GMP), 2-Hydroxybutyric acid, Linoleoyl ethanolamide, N-Acetylaspartic acid, Trehalose, Xanthosine 5’-phosphate (XMP), Guanosine diphosphate (GDP), Cytidine triphosphate, Guanosine-5’-tridiphosphate (GTP), 3-Phosphoglycerate, and Lactose.

These differently abundant metabolites enriched some important functions (Figure 4). One function, metabolic pathways, was enriched by all pairs of the four groups (*p* < 0.05). Other than that, the enriched functions were different. For example, different pathways were enriched in different groups with differently abundant metabolites. Specifically, ABC transporters (*p* < 0.04) and Biosynthesis of amino acids (*p* = 0.05) were significant between MT1 vs. WT at t2. Additionally, some groups were marginally enriched; for example, amino sugar and nucleotide sugar metabolism in WT, t2 vs. t1 (*p* = 0.06) (Figure 4A); fatty acid metabolism in MT1 vs. WT at t2 (*p* = 0.07) (Figure 4D); and the metabolism of various amino acids in MT1 vs. WT at t1 and MT1 vs. WT at t2 (*p* = 0.08) (Figure 4B,D).

### 3.3. Transcriptional Changes in B. subtilis between Four CONDITIONS

We also examined the gene expression profiles of the WT and MT1 at both timepoints. About one-quarter to one-third of the total number of genes were differentially expressed: 1744 (39.2%; MT vs. WT at t1), 1425 (32.1%; MT vs. WT at t2), 1133 (25.5%; WT, t2 vs. t1), and 1854 (41.7%; MT, t2 vs. t1). Most of the functional groups enriched by these differentially expressed genes were different between WT and MT1 in terms of KEGG pathways (Figure 5). Interestingly, most of the upregulated functional groups were for metabolism and biosynthesis. A total of 6 out of 27 KEGG pathways were shared by upregulated genes between t2 and t1 (Figure 5A), for the biosynthesis of purine (bsu00230) and pyrimidine (bsu00240) and many amino acids’ metabolism (bsu00220:arginine biosynthesis). Specifically, the identities of amino acids having upregulated expression were different between WT and MT1. Expectedly, the functional groups enriched by downregulated genes (Figure 5B) were all different from the upregulated functional groups. For example, the downregulated groups in WT between t1 and t2 included many metabolisms and biosyntheses of metabolites not present in the upregulated groups, such as pyruvate metabolism, biotin metabolism, or fatty acid biosynthesis. Similar patterns were also observed in the other groups. Specifically, the groups bsu00920:sulfur metabolism, bsu00340:histidine metabolism, and bsu02010:ABC transporters were enriched by the downregulated genes but not by the upregulated genes.

### 3.4. Integrated Transcriptomic and Metabolomic Responses to Genetic and Environmental Perturbations

We explored the integrated responses of *B. subtilis* to genetic and environmental perturbations by combining transcriptomes and metabolomes in three ways. First, we examined the similarity of the overall patterns of the transcriptomes and metabolomes through PCA. Interestingly, the PCA patterns of the top two PC (principal components) were similar among the four groups based on transcriptomes (Figure 6A) or metabolomes (Figure 6B), i.e., the three samples of the same groups were clustered together but were separated from other clusters of samples.

Next, we examined the relationship through the correlation between the metabolomes and transcriptomes. Four major clusters of correlation were identified, with each cluster of genes showing different patterns of correlation between genes and identified metabolites (Figure 6C). Specifically, in Cluster 1 and 3, most genes displayed a positive correlation with some metabolites and a negative correlation with other metabolites at the same time. In Cluster 2 and 4, there were small proportions of genes showing significant correlations which were either positive or negative instead of being correlated both ways. We also examined the biological processes enriched by the genes in each cluster. As shown in Figure 6D, each cluster had its unique functions in terms of KEGG pathways. For example, the genes in Cluster 1 were mainly involved in biosynthesis, including the biosynthesis of amino acid (particularly arginine) and sugar or sugar units, sulfur metabolism, and ABC transport. Cluster 3 had enriched functions such as cofactor and secondary factor biosynthesis and carbon metabolism. Cluster 2 was small and, therefore, had no enriched functions, while Cluster 4 had only two enriched functions: ABC transport and microbial metabolism in a diverse environment.

Last, we examined the relationship between transcriptomes and metabolomes using co-occurrence network analysis implemented with WGCNA. In contrast to the above direct correlations between individual genes and individual metabolites, the relationship between a group (module) of ‘co-regulated’ genes and individual metabolites was identified. A total of 10 modules of genes showed different correlations with metabolites (Figure 6E). The genes in each module showed different correlations with metabolites. Additionally, we also examined the biological processes of each module of enriched genes. As two examples, only two big modules (MEblue and MEturquiose) were used to enrich the functions. Each module enriched their distinct functions in terms of KEGG pathways. Specifically, module MEblue enriched four functions, mainly the metabolism of pyrimidine and cysteine and methionine, ABC transport, and beta-lactam resistance. The module MEturquioise enriched functions in secondary and cofactor metabolism, amino acid biosynthesis, the pentose phosphate pathway, carbon metabolism, and some special metabolisms in diverse environments.

## 4. Discussion

This study investigated the integrated response of a GlcNAc-producing *B. subtilis* strain to internal (genetic) and external (environmental) perturbations, by combining transcriptomics and metabolomics. Interestingly, these two omics approaches revealed the same pattern of responses in terms of biological processes. Importantly, these integrated responses were different between the four groups of growth conditions.

*B. subtilis* has been a model organism as a cell factory [14]. GlcNAc yield is another important feature of the engineered strain MT1. In parallel to its smaller maximum growth rates than the wildtype, to support its higher carrying capacity, there were higher levels of energy supply in terms of ATP and GTP, with the energy state metric differing among the four groups. Another piece of evidence for sufficient energy supply was the low NAD^+^/NADH ratios, which were between 0.22 (WT) and 0.42 (MT1) at time 1. Additionally, in the engineered strain MT1, most of the key precursors were also higher at time 2 than time 1, a sign of increased supply, such as D-ribose 5-phosphate and glucose 6-phosphate in central metabolism [15]. Meanwhile, the lower acetyl-CoA and glutamine may suggest the fast drain of these key metabolites into biosynthesis [16]. Additionally, for 13 key precursor metabolites, there were some of them which were always differently abundant between any of the four groups, including GTP, CTP, 5′-Deoxyadenosine, Adenosine 3′,5′-diphosphate, Guanosine monophosphate (GMP), 3-Phosphoglycerate, Lactose, etc. Some of them are energy or precursor metabolites. These profiles of the concentrations fit the three topological subnetworks well: catabolism, core, and anabolism [17].

Besides being a cell factory, *B. subtilis* has been a model organism for systems biology research [18]. Using the strain with an introduced gene GNA1 (encoding GlcNAc-6-phosphate N-acetyltransferase) from *S. cerevisiae* S288C, it is treated with both genetic (GNA1 introduction) and environment (carbon source: glucose vs. glucose + xylose, i.e., t1 vs. t2 in this study) perturbations. This led to the high yield of GlcNAc in this study as well as reported previously [7]. This engineered strain serves as a perfect model system to address the integration of multiomics in response to perturbation. Both substantial transcriptional and metabolomic changes were observed between the four groups. The integration of two omics was studied in five types of analyses, namely, gene expression and metabolite abundance in central metabolism, the PCA clustering of groups separately by metabolomics and transcriptomics, the individual genes which show strong (positive or negative) correlations with metabolites, the module of genes which are coexpressed, and the biological functions these genes enrich in terms of KEGG pathways. Importantly, these integrated analyses revealed the same patterns of response. Among the key pathways underlying different responses, prominent functions include the biosynthesis of amino acids and of sugar or sugar units, sulfur metabolism, ABC transport, cofactor and secondary factor biosynthesis, and carbon metabolism. Among them, ABC transport and secondary metabolites are important to amino acid transport and metabolism. Amino acids are the building blocks of proteins; the changes indicate they are needed for growth. The costs of synthesis are between 12 and 74 high-energy phosphate bonds per molecule in *B. subtilis*, so there are less energetically costly amino acids in highly expressed proteins, which is a result of natural selection to maintain metabolic efficiency [19]. This high level of consistency between omics has been observed in *B. subtilis* under other growth conditions. For example, in a multiomics study of the *B. subtilis* strain BJ3-2 cultured at 37 °C, 45 °C, and 53 °C, transcriptional changes related to secondary metabolites and ABC transporters were increased, as in our study; more interestingly, proteomics and metabolomics profiling also confirmed the marked change in amino acid transport and metabolism [4]. More interestingly, the metabolic differences discovered in our study (e.g., sulfur metabolism, cysteine and methionine metabolism, and pyrimidine metabolism) were also observed between 45 °C and 53 °C compared to 37 °C. In another multiomics study of *B. subtilis* responses to simultaneous nutrient limitation and osmotic stress, a major response under osmotic stress is the production of proline (compatible solute) through the concerted adjustment of multiple reactions around the 2-oxoglutarate [20]. 

Another major finding is the high consistency between two omics approaches. First, both metabolomes and transcriptomes revealed the same PCA clusters of the four groups. Second, we found that the important functions enriched both by metabolomes and transcriptomes overlapped, such as amino acid metabolism and ABC transport. Strikingly, these functions overlapped those enriched by the genes showing strong (positive or negative) correlation with metabolites. All these findings suggest that the responses to genetic and environmental perturbations are well coordinated at the metabolic and transcriptional levels. On a higher level, this consistency reflects the highly hierarchical modular organization of the genome-scale metabolic network [21] and the dynamic coordination of multiomics in real operation conditions [17]. 

## Figures and Tables

**Figure 1 microorganisms-11-01949-f001:**
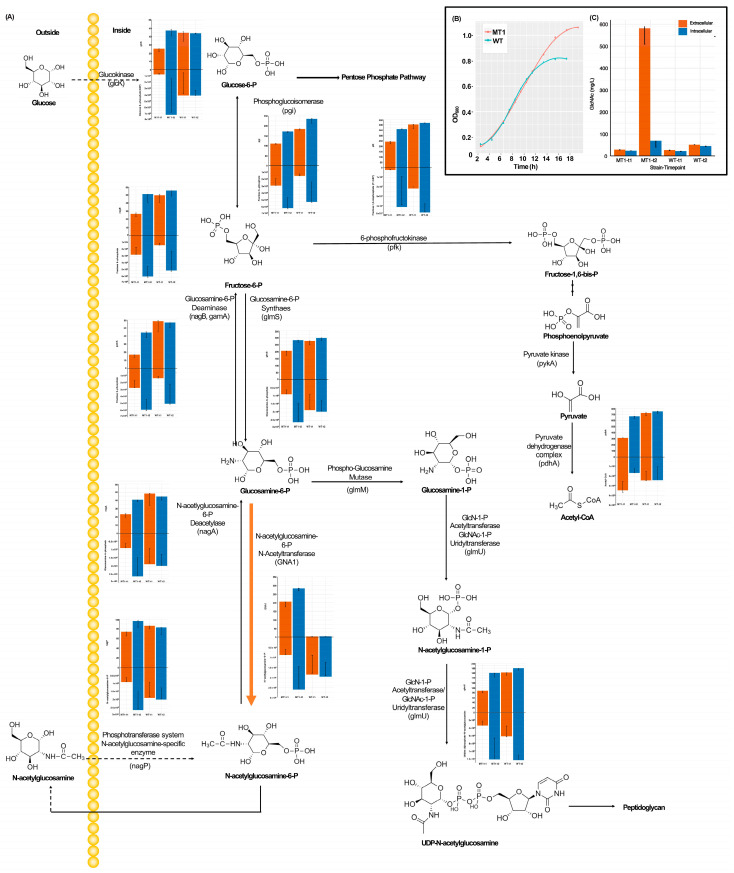
The growth curve and GlcNAc production under aerobic conditions, along with gene expression levels and metabolite concentrations in the associated pathways. (**A**) Gene expression levels and metabolite concentrations. The bar plots facing up are gene levels and those facing down are me-tabolite concentrations. The orange arrow indicates introduced GNA1 gene. (**B**) The growth curve of WT and MT1. (**C**) Intracellular and extracellular GlcNAc production. The x-axis represents the four groups (from left to right, t for timepoint): MT1-t1, MT-t2, WT-t1, and WT-t2.

**Figure 2 microorganisms-11-01949-f002:**
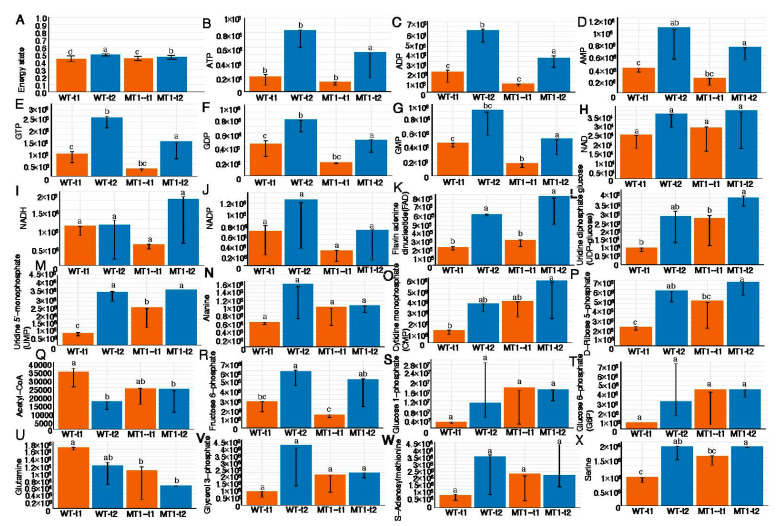
The key metabolite indicators for cell metabolic state in terms of energetics, redox, and essential precursors. Multiple comparison methods are used to label the differences.

**Figure 3 microorganisms-11-01949-f003:**
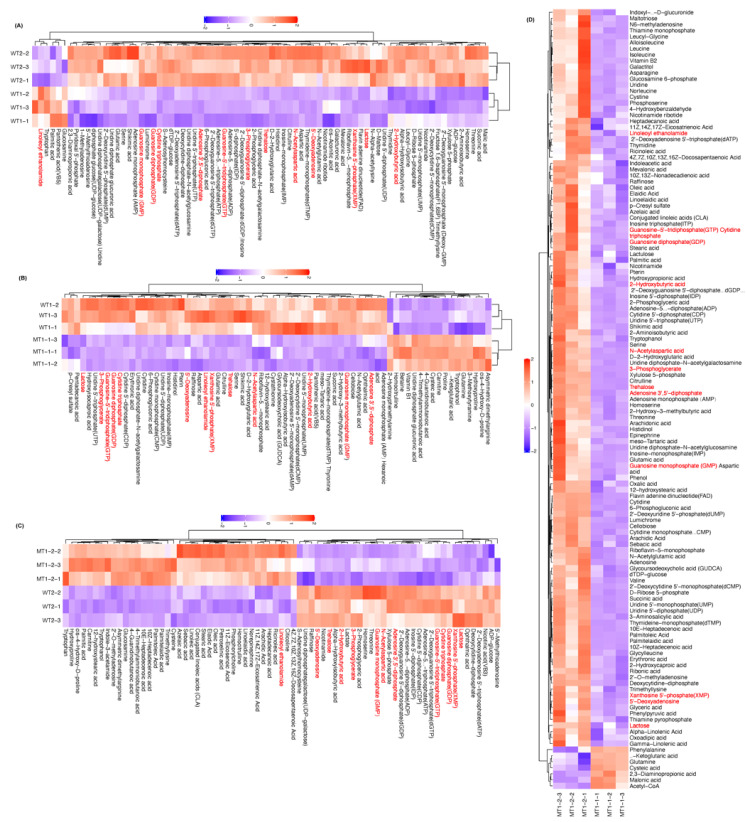
The metabolites showing different abundance between the groups. (**A**): WT−t2 vs WT−t1. (**B**): MT1−t1 vs WT−t1; (**C**): MT1−t2 vs WT−t2; and (**D**): MT1−t2 vs MT1−t1.

**Figure 4 microorganisms-11-01949-f004:**
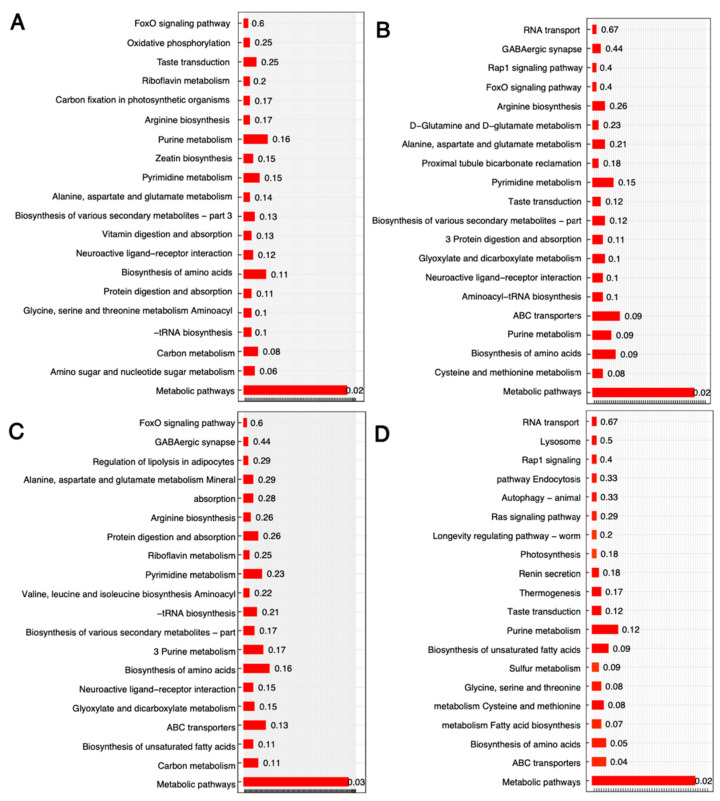
The KEGG pathways enriched by differentially abundant metabolites between different groups. (**A**): WT-t2 vs. WT-t1; (**B**): MT1-t1 vs. WT-t1; (**C**): MT1-t2 vs. MT1-t1; and (**D**): MT1-t2 vs. WT-t2.

**Figure 5 microorganisms-11-01949-f005:**
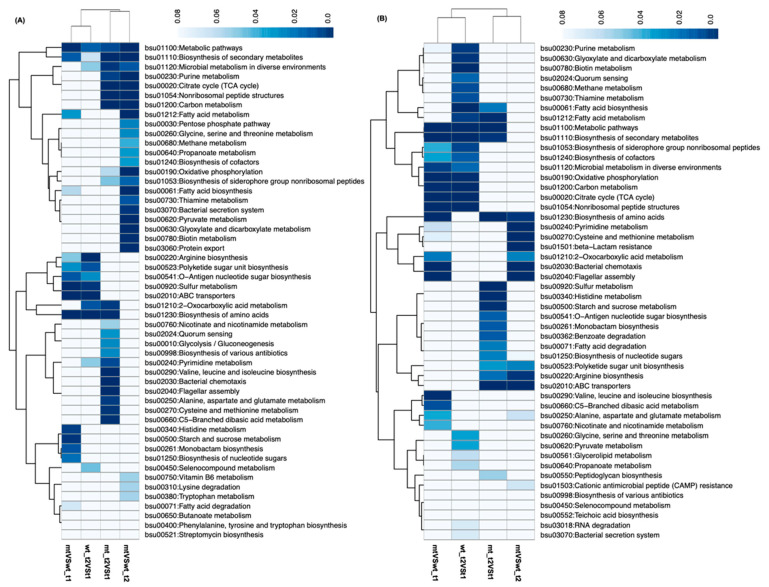
The KEGG terms enriched by up- and downregulated genes. (**A**): Terms enriched by upregulated genes and (**B**): terms enriched by downregulated genes.

**Figure 6 microorganisms-11-01949-f006:**
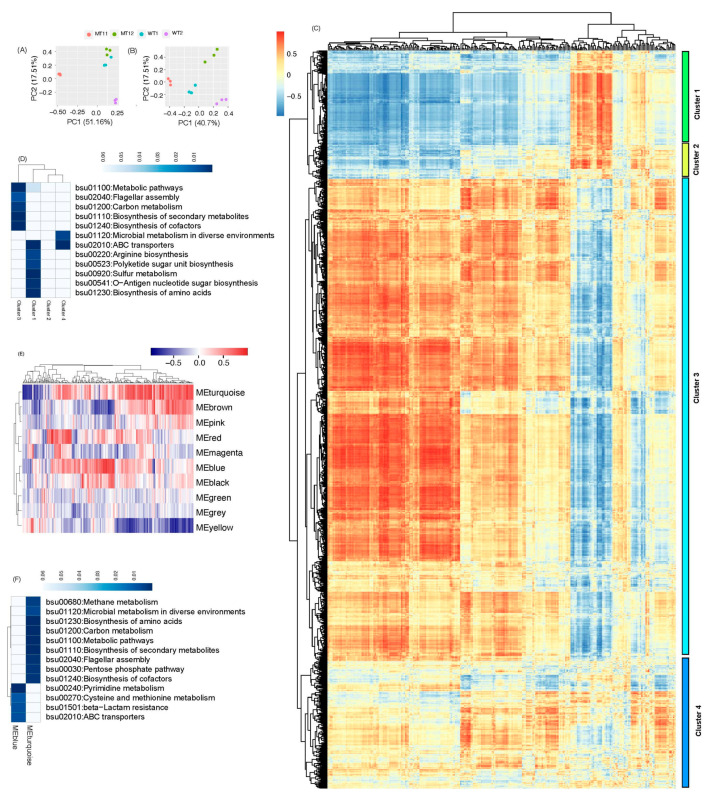
The integration of transcriptome and metabolome in *B. subtilis* under four growth conditions. (**A**) PCA clusters of the four groups based on the metabolomes; (**B**) PCA cluster of the four groups based on transcriptomes; (**C**) the correlation between metabolite abundance and gene expression level. Rows are genes and columns are metabolites; (**D**) the KEGG pathways enriched by the genes in the different metabolite−gene correlation clusters; (**E**) the modules (subnetworks) of the genes revealed through network analysis; and (**F**) the KEGG pathways enriched by the genes in two big modules from network analysis.

**Table 1 microorganisms-11-01949-t001:** The *Bacillus subtilis* strains used and recreated in this study.

Strains	Characteristics	Reference
WT: *B. subtilis* 168	*Bacillus subtilis* subsp. *subtilis* str. 168	Lab stock
MT1: BNZR0.00/pP43-GNA1	*Bacillus subtilis* 168 derivate, epr::PxylA-comK, trpC0, aprE::gamR-Psg2-lacO-RBS4-T7rnap-PT7gam-egfp, pP43-GNA1	This work
MT2: BNZR1.06/pP43-GNA1	BNZR0.00 derivate, ∆gamA∆nagABP∆ldh∆ackA, PlysC-glmS, pP43-GNA1	This work

## Data Availability

The data is available at NCBI under the BioProject accession number PRJNA954209. The dataset can be accessed at the following URL: https://www.ncbi.nlm.nih.gov/bioproject/PRJNA954209 (accessed on 26 July 2023).

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
