# Peer review of "The Multiomics Response of Bacillus subtilis to Simultaneous Genetic and Environmental Perturbations"

_microorganisms, 2023, doi:10.3390/microorganisms11081949_

Round 1
Reviewer 1 Report
The manuscript under review concerns an important issue from the general biological point of view, which is the reaction of organisms to genetic and environmental disorders. The above issue is an important problem for both scientists and practitioners, for example crop breeders. The authors of the reviewed manuscript used the common bacterium Bacillus subtilis as a model to understand respond at the systems level to both genetic and environmental perturbations. As the authors write, bioengineering or synthetic biology provides an ideal system for studying such responses, as the engineered strains always have genetic changes as opposed to wildtypes and are grown in conditions which often change during growth for maximum yield of desired products. The idea behind the authors was to investigate how a strain of Bacillus subtilis with overproduction of N-acetylglucosamine, which were grown in an environment with carbon shift from glucose to glucose and xylose, would react in the same culture system. Four combinations of both wild and mutant bacteria and different combinations of carbon source were used in the experiment. After the incubation period, detailed genetic studies measuring transcriptomes and metabolomes began. It should be noted that detailed breeding and genetic studies were carried out in accordance with generally accepted methods commonly used in laboratories. This study consisted of a detailed analysis by sequencing the genome of bacteria that grew in artificially created conditions that were carefully controlled. The correlation between transcriptome and metabolome was analyzed using two approaches. First, direct correlations between them were calculated using the R base function cor. The genes showing high correlations with metabolomes were extracted to enrich GO and KEGG terms in the webserver DAVID. Co-occurrence network analysis was performed using R package WGCNA to identify the groups of the genes associated with metabolites. The modules (co-expressed subnetworks) were also used to enrich GO and KEGG biological functions using DAVIDe. The results obtained were thoroughly characterized and described and presented in 6 colored figures, which clearly indicate that these two omics approaches revealed the same pattern of responses in terms of biological processes and that these integrated responses were different between the four groups of growth conditions. In addition, this engineered strain serves as a perfect model system to address the integration of multi-omics in response to perturbation. Both substantial transcriptional and metabolomic changes were observed between the four groups. The authors showed that major finding is the high consistency between two omics approaches. First, both metabolomes and transcriptomes revealed the same PCA clusters of the four groups. Second, that the important functions enriched both by metabolomes and transcriptomes overlap, such as amino acid metabolism and ABC transport. Strikingly, these overlap functions those enriched by the genes showing high (positive or negative) correlation with metabolites. All these findings suggest that the responses to genetic and environmental perturbations are well coordinated at the metabolic and transcriptional levels. On a higher level, this consistency reflects the highly hierarchical modular organization of the genome-scale metabolic network and the dynamics coordination of multi-omics in real operation. In the paper, the authors cite 21 items of the latest publications, which are well selected and used to compare the results obtained. The results and their discussion do not raise any scientific or substantive objections. Considering the scientific significance of the conducted research and its practical aspect, which can be used in breeding work on cultivated plants, I believe that the evaluated work should be published unchanged.
Author Response
Thank you so much for your insight and affirmation!
Reviewer 2 Report
Dear authors,
The manuscript about multi-omics of Bacillus subtilis showed the differences between a WT and a mutant to genetic and environmental modifications. The results showed important pathways and correlations between the different comics that were used. However, some information is unclear which makes it difficult to understand the manuscript.
I would suggest to rewrite the methodology on the strain conditions. In the abstract 4 groups are highlighted but this in not clear in section 2. Also in these 4 groups MT2 is not referred which than later on appears in some results. I understand that t1 is from glucose only and t2 from glucose and xylose, but again this is not very easy to understand for the reader from section 2.3. It seems that t1 comes from a third cycle after adding xylose at OD600 0.4, although this should not be the case. Please reformulate the sentences, and separate the conditions clearly.
I could not find figure 1C and 4E
Also is Fig.S2 correct? The manuscript refers to metabolites but GlcNAc production is shown.
Please keep the same definitions throughout the manuscript, timepoint, t1 or time 1?
The project accession number gives me a different project, please check.
Methods: section 2.2, write GlnAc in full one time first
Please change sentences written with "we use" or "we did", it should be for example, engineered strains were used, or four groups were used.
In the abstract, close to the end of the section, "furthermore , these sections also overlap (delete enriched). The last sentence in the abstract "of synthetic".
In the introduction "...to the factor of external environment but not for the genetic factor as the test organisms..." what do authors mean? should it be of external environment but not for the genetic factors to test if organisms are?
Author Response
Dear authors,
The manuscript about multi-omics of Bacillus subtilis showed the differences between a WT and a mutant to genetic and environmental modifications. The results showed important pathways and correlations between the different comics that were used. However, some information is unclear which makes it difficult to understand the manuscript.
I would suggest to rewrite the methodology on the strain conditions. In the abstract 4 groups are highlighted but this in not clear in section 2. Also in these 4 groups MT2 is not referred which than later on appears in some results. I understand that t1 is from glucose only and t2 from glucose and xylose, but again this is not very easy to understand for the reader from section 2.3. It seems that t1 comes from a third cycle after adding xylose at OD600 0.4, although this should not be the case. Please reformulate the sentences, and separate the conditions clearly.
Response: Great points. As suggested, we added the four groups in the Methods 2.3 section. We also referred to MT2 in the Methods (Table 1) and Results. We did restate the groups to distinguish the conditions clearly. For growth cycles in minimal medium, there were two acclimation cycles and samples were collected at the two timepoints in the third growth cycle, in which cells first grew on glucose and then glucose and xylose.
I could not find figure 1C and 4E
Response: For Figure 1C, it is in the upper right corner of Figure 1, alongside Figure 1B. For Figure 4E, it should be Figure 6E.
Also is Fig.S2 correct? The manuscript refers to metabolites but GlcNAc production is shown.
Response: Fig. S2 should be Fig. 2.
Please keep the same definitions throughout the manuscript, timepoint, t1 or time 1?
Response: As suggested, we used t1 and t2 throughout the revised text for consistency and clarity.
The project accession number gives me a different project, please check.
Response: The correct project accession number is PRJNA954209.
Methods: section 2.2, write GlnAc in full one time first
Response: For methods: Section 2.2, GlnAc was first written in its full name on its first appearance.
Comments on the Quality of English Language
Please change sentences written with "we use" or "we did", it should be for example, engineered strains were used, or four groups were used.
In the abstract, close to the end of the section, "furthermore , these sections also overlap (delete enriched). The last sentence in the abstract "of synthetic".
In the introduction "...to the factor of external environment but not for the genetic factor as the test organisms..." what do authors mean? should it be of external environment but not for the genetic factors to test if organisms are?
Response: Great points! We corrected the statements as suggested and the similar problems elsewhere in the manuscript.
Round 2
Reviewer 2 Report
Dear authors
Thank you for the revision of your manuscript, which improved.
A small point, as Fig. 2S meant to be Fig. 2 as stated by the authors and corrected, where is the reference to Fig.2S now then?
Author Response
Dear authors
Thank you for the revision of your manuscript, which improved.
A small point, as Fig. 2S meant to be Fig. 2 as stated by the authors and corrected, where is the reference to Fig.2S now then?
Response: Fig. S2 is the production of GlcNAc in MT2 under aerobic conditions. However, since MT2 was not the main subject of our study. We therefore did not include the results of this figure in the main text. It is only included as an additional experiment in the supplement figure